

# Thaw processes in ice-rich permafrost landscapes represented with laterally coupled tiles in a Land Surface Model

Kjetil S. Aas[1], Léo Martin[1], Jan Nitzbon[1,2,3], Moritz Langer[2,3], Julia Boike[2,3], Hanna Lee[4], Terje K. Berntsen[1], Sebastian Westermann[1]

[1] University of Oslo, Department of Geosciences, Sem Sælands vei 1, 0316 Oslo, Norway
[2] Alfred Wegener Institute, Helmholtz Centre for Polar and Marine Research, Telegrafenberg A45, 14473 Potsdam, Germany
[3] Humboldt University of Berlin, Geography Department, Unter den Linden 6, 10099 Berlin, Germany
[4] NORCE Norwegian Research Centre, Bjerknes Centre for Climate Research, Jahnebakken 5, 5007 Bergen, Norway

*Correspondence to*: Kjetil S. Aas (k.s.aas@geo.uio.no)

**Abstract.** Earth System Models (ESMs) are our primary tool for projecting future climate change, but are currently limited in their ability to represent small-scale land-surface processes. This is especially the case for permafrost landscapes, where melting of excess ground ice and subsequent subsidence affect lateral processes which can substantially alter soil conditions and fluxes of heat, water and carbon to the atmosphere. Here we demonstrate how dynamically changing microtopography and related lateral fluxes of snow, water and heat can be represented with a tiling approach suitable for implementation in large-scale models, and investigate which of these lateral processes are important to reproduce observed landscape evolution. Combining existing methods for representing excess ground ice, snow redistribution and lateral water and energy fluxes in two coupled tiles, we show how the same model approach can simulate known degradation processes in two very different kinds of permafrost landscapes. Applied to polygonal tundra in the cold, continuous permafrost zone, we are able to simulate the transition from low-centered to high-centered polygons, and show how this results in i) more realistic representation of soil conditions through drying of elevated features and wetting of lowered features with related changes in energy fluxes, ii) reduced average permafrost temperatures at 13 m depth with up to 2 °C in current (2000-2009) climate, iii) delayed permafrost degradation in the future RCP4.5 scenario by several decades, and iv) more rapid degradation through snow and soil water feedback mechanisms once subsidence starts. Applied to warm, sporadic permafrost features, this two-tile system can represent an elevated peat plateau underlain by permafrost in a surrounding permafrost-free fen, and how it degrades in the future following a moderate warming scenario. These results show the importance of representing lateral fluxes to realistically simulate both the current permafrost state and its degradation trajectories as the climate continues to warm; both of which are likely to have important implications for simulations of the magnitude and timing of the permafrost carbon feedback.



# 1 Introduction

Permafrost landscapes represent an important, but complex component of the Earth's climate system. They currently cover approximately one quarter of the land area in the Northern Hemisphere (Zhang et al., 1999), and exert a major control on the local and regional hydrology and ecology. Moreover, it is estimated that approximately 1300 Pg carbon is stored in this region, which is considerably more than the current atmospheric carbon pool (Hugelius et al., 2014). If thawed and mobilized, this carbon could become a major source of greenhouse gas emissions (Schuur et al., 2008). However, continued high-latitude warming and widespread permafrost thaw will likely also be associated with large-scale vegetation changes, which could act as an important carbon sink (Qian et al., 2010, McGuire et al. 2018). Understanding the future evolution of permafrost landscapes, and associated changes in the biogeochemical cycles, is therefore important for future estimates of climate change (Schuur et al., 2015).

Our primary tool for estimating future climate change, including the magnitude and interplay between related climate feedbacks is the comprehensive Earth system models (ESMs). Due to the possibly large impact of the permafrost-carbon feedback (PCF) on the climate system, permafrost processes have received much attention in the development of these models during the last decade. Considerable improvements have been made by including freeze-thaw processes, multilayer soil carbon representation, increased soil depth and resolution, moss representation and multilayer snow schemes (Lawrence and Slater, 2005; Koven et al., 2013b; Chadburn et al., 2015; Burke et al., 2013). However, a major limitation remains in the lack of representation of subgrid-scale permafrost processes in these models (Lawrence et al., 2012; Beer, 2016). In particular, the ability to simulate changing microtopography resulting from melting of excess ground ice (therokarst) is lacking. These processes are currently observed many places in the Arctic. In polygonal tundra, Liljedahl et al. (2016) have documented how low- and flat-centered polygons (LCP and FCP) are transitioning into high-centered polygons (HCP), with large associated changes in local and regional hydrology. On the other hand, sporadic or isolated permafrost features like palsas and peat plateaus can be maintained only through small-scale elevation differences and lateral fluxes of snow and water (Seppälä, 2011). Melting of excess ice in these features sets off a feedback mechanism through subsidence, enhanced snow accumulation, reduced winter heat loss and increased soil ice melt, which cannot be represented in a single large-scale grid cell. Accounting for these processes in ESMs is of particular importance since the regions with high amounts of excess ice are to a large degree also regions with high amounts of soil carbon. Olefeldt et al. (2016) estimated that 20% of the northern permafrost region is covered by thermokarst landscapes, but suggested that as much as 50% of the soil organic carbon (SOC) in this region could be stored here.

The challenges with capturing the hydrologic response of degrading permafrost have been described by Painter et al. (2013), who partitioned these processes into "subsurface thermal/hydrology, surface thermal processes, mechanical deformation and overland flow processes". Some of these have been addressed in individual studies on local scales. For instance, polygonal tundra in Alaska has been simulated by Kumar et al. (2016) using a multiphase, 3D thermal hydrology model (PFLOPTAN), by Grant et al. (2017) who included lateral fluxes of subsurface water as well as redistribution of snow





and surface water, and by Bisht et al. (2018) who simulated a 104 m long transect with sub-meter resolution including snow redistribution and lateral water and energy fluxes. In the warmer (discontinuous and sporadic) permafrost zones, Kurylyk et al. (2016) and Sjöberg et al. (2016) included groundwater flow and related heat advection in the simulation of peat plateaus in Canada and Sweden, respectively. Although capturing different aspects of lateral fluxes in ice-rich permafrost landscapes,

these simulations have been performed with models running on high-resolution grids which are not transferable to largescale Land Surface Models (LSMs). Nor have they included the mechanical deformation aspect needed to represent transient landscape changes, or treated this in a unified way that can be applied to both continuous and discontinuous/sporadic permafrost features.

On the larger scale, Lee et al. (2014) included excess ground ice in a global LSM simulation which estimated land

subsidence related to permafrost thaw and ground ice melt, but without including subgrid-scale variations and related lateral fluxes. Qui et al. (2018) included a separate subgrid tile in an LSM receiving surface runnof from the surrounding tiles to simulate peatlands with related carbon, moisture and energy fluxes. Gisnås et al. (2016) and Aas et al. (2017) used subgrid tiles to represent heterogeneous snow accumulation, and showed how this influenced soil thermal regime and surface energy fluxes, respectively. Finally, Langer et al. (2016) employed a two-tile approach to simulate lateral heat exchange in polygonal

tundra, to show that heat loss to surrounding land masses was needed to simulate stable thermokarst lakes in Northern Siberia.

Here we extend the two-tile approach of Langer et al. (2016) with lateral fluxes of snow and subsurface water flow, and combine this with the excess ice formulation of Lee et al. (2014). In this way, we simulate changing microtopography dynamically together with the effect this has on lateral fluxes of snow, water and heat. We thereby aim to for the first time simulate dynamical landscape changes due to excess ice melt, and related changes in lateral fluxes, in a framework suitable

for implementation in ESMs. We apply this laterally coupled two-tile system to a polygonal tundra site in Northern Siberia and a peat plateau location in Northern Norway, and compare with results from a standard 1-D reference simulation. The two sites represent cold, continuous permafrost and warm, sporadic permafrost, respectively. Signs of permafrost degradation can currently be found at both locations, and small-scale heterogeneity in soil moisture and snow accumulation is a common feature for the two locations. Hence, they represent two very different climatic conditions where current large-scale models fail to

capture key small-scale processes that are important for the soil thermal regime. By testing the model at these two locations we show to what extent the same simple model approach can represent known landscape changes and related water and energy fluxes under very different permafrost conditions, and evaluate which of the lateral fluxes are important at the two locations. Capturing the detailed properties at the specific test sites is not the objective of this study, but rather to show how the general behavior observed in these landscapes can be represented in a way suited for large-scale models.



## 2 Methods

### 2.1 Site descriptions

The model is applied to the two permafrost locations shown in Fig. 1. Samoylov Island in northern Siberia represents a
polygonal tundra location in cold, continuous permafrost, whereas the peat plateaus in Suossjavri, northern Norway, represents
warm, sporadic permafrost. Both locations are, however, examples of carbon- and ice-rich permafrost landscapes where small-
scale lateral fluxes are important for representing the physical state of permafrost.

### 2.1.1 Samoylov Island, Northern Siberia

Samoylov Island (72º22'N, 126º28'E) is located in the southeast corner of the Lena River delta. The size of the entire
delta including more than1500 islands and about 60 000 lakes is about 25 000 km² (Fedorova et al., 2015) and underlain by
continuous, cold permafrost. The island of Samoylov, located in the southern part of the delta, mainly consists of polygonal
tundra surrounding a number of ponds and lakes (Fig. 1b; Boike et al., 2013, 2018). All degradation stages described by
Liljedahl et al. (2016) can be found here, from non-degraded low-centered polygons to high-centered polygons with connected
troughs (see Nitzbon et al., 2018). Between 1997 and 2017 the mean annual air temperature at the island was approximately -
12.3 °C, with an annual liquid precipitation of 169 mm and mean end-of-winter snow depth of 0.3 m (Boike et al., 2018). The
depth of zero annual amplitude is at 20.8 m, and has warmed from - 9.1 °C in 2006 to - 7.7 °C in 2017. Numerous studies have
been conducted on the island, including studies of water and surface energy balance (Boike et al., 2008, Langer et al., 2011a,
b) and carbon cycling (Knoblauch et al., 2018; Knoblauch et al., 2015). As a well-studied site with available meteorological,
soil physical, and hydrological measurements (Boike et al., 2018), it has also been used as test site for various permafrost
modelling studies, including ESM validation and development (Chadburn et al., 2015; Chadburn et al., 2017; Ekici et al., 2014;
Ekici et al., 2015).

### 2.1.2 Suossjavri, Northern Norway

Suossjavri (69º23'N, 24º15'E) is situated in the central part of Finnmark county in northern Norway. It is part of the sporadic
permafrost zone in northern Fennoscandia (Fig. 1), where permafrost outside mountain regions is confined to palsas and peat
plateaus in mires. The site has an elevation of approximately 335 m asl and covers over c.a. 23 ha. It is bordered by the Iesjoka
River on the South and the Suossjavri Lake on the East and North and consists of metric to decametric palsas and peat plateaus
that rises 20 cm to 2 m above the seven surrounding wet mires. These permafrost bearing morphologies are currently degrading,
and have lost approximately 30 % of their area in the last six decades (Borge et al., 2017). Largest degradation rates are seen
for the smaller plasas and peat plateaus, which have lost almost half their areas in this period, compared to only 15%
degradation of the four largest peat plateaus.
30           The mean annual air temperature in the region is usually comprised between -2°C and -4°C with a summer mean
value of 10°C (JJA) and winter value of -15°C (DJF, Aune, 1993, for the 1961-1990 period). The mean annual precipitation





is below 400 mm according to the nearest measurement station (Borge et al., 2017). Mean annual ground surface temperatures (MAGST) have been measured with iButton® temperature loggers at 25 locations across one of the mires since September 2015, along with end-of-summer thaw depths and end-of-winter snow depths at the same points. These show snow depths on the interior peat plateaus mostly between zero and 40 cm, ALT between 40 cm and 70 cm, and 1°C to 2°C colder MAGST

than the surrounding mires.

## 2.2 The NoahMP land surface model

Our modeling study is performed with the NoahMP LSM version 3.9, with a number of modifications described below. In its default configuration the NoahMP model (Niu et al., 2011) simulates soil temperature and frozen and liquid water in four soil layers down to a depth of 2 m. It includes up to three snow layers with representation of liquid water retention and refreezing,

as well as a separate canopy layer. Compared to the original Noah code, NoahMP is an augmented version that includes multiple alternative model representations for key processes, including parameterizations of supercooled liquid water and frozen ground hydraulic conductivity (see details in Niu et al., 2011). It is substantially less complex and computationally expensive than LSMs used in current state-of-the-art ESMs, disregarding for instance biogeochemical processes and dynamical vegetation. However, in its basic treatment of soil thermal and hydrological processes, it is comparable to, and includes some

of the same parameterizations as, the Community Land Model (CLM; Lawrence et al., 2011). In the following, we will describe the modifications and augmentations to the NoahMP model for our simulations.

### 2.2.1 Soil resolution, excess ice and soil organic fraction

To better represent permafrost processes, the number of soil layers was increased from the default four to 37, with the total soil depth increased from 2 m to 7 m or 14 m, plus excess ice thickness (Fig. A1). These soil depths where chosen to

approximately include the zero annual amplitude depth at Suossjvari and Samoylov, respectively, but still be shallow enough to avoid long spin-up times as the emphasis here is on the near surface processes rather than the deep soil conditions.

Second, we added soil organic fraction as an additional (fixed) input variable. Following Lawrence and Slater (2008), soil thermal and hydraulic properties were calculated assuming a linear weight between organic and the (original) mineral fractions. This allowed us to simulate organic rich soils like peat, which has properties very different than the default soil types

available in NoahMP.

Following Lee et al. (2014) we include excess ice within the existing layers in the model, so that the layer thicknesses and properties of the layers change throughout the simulation as the excess ice melt. Excess ice is initialized as a certain fraction (Fexice), within a certain depth region in each soil column (see Fig. 3 and A1). Because excess ice is incorporated as an initial condition, it only melts and does not grow. The water from melting excess ice is added to the soil column in the layer

where it melts, or the nearest unsaturated layer above if this layer is saturated.





### 2.2.2 Implementation of interacting tiles

Sub-grid tiles have been implemented in the original Noah version as part of the Weather Research and Forecasting model (WRF) to represent a mosaic of land cover types (Li et al., 2013). This tiling included soil columns simulated independently for each tile, but without any interaction between the tiles during the simulation. Here we build upon this methodology to

explicitly simulate individual land units within a grid cell, but include also lateral fluxes as described below. In the following general description of the interactive tiles, we will refer to these as tile 1 and 2, but later refer to them as RIM and CENTER for the polygonal tundra and PPLAT and MIRE for the peat plateau setting (Fig. 2).

### 2.2.3 Snow redistribution between interactive tiles

To represent the effect of snow redistribution by wind, we scale the amount of snow received in tile 1 and 2 based on the

difference in elevation at the top of the snow/soil column. Similar to Aas et al. (2017), this is done with a scaling factor, so that the accumulation of snow in tile $i$ is calculated according to the grid-cell mean snowfall S, times the scaling factor $f_i$ ($S_i = f_i * S$).

The scaling factor is calculated as follows. For snow depths below a minimum snow value ($Hs_{min}$), no redistribution takes place.

$$f_i = 1.0, \ for \ Hs_i < Hs_{min}, \tag{1}$$

Once the tile with the highest elevation reaches the minimum snow value, the scaling factor is calculated so that no new snow accumulates on this tile before the total snow and soil elevation ($z_i$) is within 5 cm of each other:

$$f_{1,2} = \begin{cases} 1.0, \ for \ |z_1 - z_2| < 0.05m \\ 0.0, \ for \ (z_{1,2} - z_{2,1}) \geq 0.05m \\ 1.0 + \frac{A_{2,1}}{A_{1,2}}, \ for \ (z_{2,1} - z_{1,2}) \geq 0.05m \end{cases}, \tag{2}$$

where $A$ refers to the area of the tile, and the subscript refers to the tile number (1 or 2).

### 2.2.4 Lateral subsurface water flux between interactive tiles

Lateral water flux is calculated similar to subsurface flow in WRF-hydro (Gochis et al., 2015), with a few modifications relevant for permafrost conditions. The flow rate [m³ s⁻¹] from a one tile to another can be calculated as

$$q = \begin{cases} -T tan(\beta)L, \ for \ \beta < 0 \\ 0 \qquad \quad , \ for \ \beta \geq 0 \end{cases}, \tag{3}$$

where $T$ is the transmissivity, $L$ is the contact length and $\beta$ is the water table slope between the tiles. $T$ is given by

$$T = \begin{cases} \frac{K_{sat} Z_B}{n} \left(1 - \frac{z_{wt}}{Z_B}\right)^n, \ for \ z_{wt} \leq Z_B \\ 0, \qquad for \ z_{wt} > Z_B \end{cases}. \tag{4}$$

Here $z_{wt}$ is the water table depth, $Z_B$ is the total soil depth, $K_{sat}$ is the saturated hydraulic conductivity and $n$ is a tunable local power law exponent determining the decay rate of $K_{sat}$ with depth.





Here we set $n = 1$, and use the depth to the minimum (highest) frost table depth ($z_{frzmin}$) instead of the full soil depth $Z_B$. Inserting $tan(\beta) = \frac{z_{wt_{1,2}} - z_{wt_{2,1}}}{D}$, where $D$ is the distance parameter, the flow rate can then be calculated as:

$$q_{1,2} = \begin{cases} -WK_{sat}\frac{z_{wt_{1,2}}-z_{wt_{2,1}}}{D}\left(z_{frzmin} - z_{wtmin}\right), \ for \ z_{frzmin} \geq z_{wtmin}, \\ 0, \ for \ z_{frzmin} < z_{wtmin} \end{cases} \qquad (5)$$

Here we set the frost table to the top of the first layer (from the top) with more than 1 % volumetric soil ice (including excess ice). The water table depth is taken as the depth to the top of the first saturated soil layer.

### 2.1.5 Lateral ground heat flux between interactive tiles

The lateral ground heat flux [W m$^{-2}$] between two grid cells with overlapping soil depth of $\Delta z$ can be calculated as (see Langer et al., 2016):

$$qs_{1,2} = \frac{L}{A_{1,2}}k_s\frac{T_{2,1}-T_{1,2}}{D}\Delta z, \qquad (6)$$

where $k_s$ is the thermal conductivity. This is calculated individually for each partially overlapping soil layer.

### 2.3 Model setup and forcing

The model setup is shown in Fig. 3 and Table 1 and 2 and described separately for the two locations in the following, together with the forcing data for the corresponding locations. In both cases, a model timestep of 15 min is applied, with zero flux as the lower thermal boundary condition. To represent larger-scale landscapes with a small number of tiles, we exploit the concept of self-similarity (i.e. translational symmetry). At both locations, a separate reference simulation (REF) is run with the same initial conditions as the elevated tile in the coupled system (RIM or PPLAT). The other (initially lower) tile in the coupled simulation is referred to as CENTER and MIRE for the tundra and mire locations, respectively.

### 2.3.1 Polygonal tundra on Samoylov Island, Northern Siberia

The polygonal landscape at Samoylov Island is here represented with two tiles that represent center regions and rim regions, respectively. These are in reality of different sizes and shapes (Fig. 1), but can to a first approximation be considered a self-repeating pattern, as also described by Nitzbon et al. (2018). Due to symmetry, a larger region can then be represented as a single feature with a representative geometry, where the interaction between the different polygons can be ignored. For simplicity, we here simulate a representative polygon as a circular feature with center and rim of equal area, and a total diameter of 10 m. Assuming instead hexagons, like Nitzbon et al. (2018), would only require minor modifications to the parameters shown in Fig. 3, particularly the distance parameter and the interaction length.

To represent an ice wedge occupying the majority of the soil volume, we initialize the RIM with an excess ice fraction of two thirds between the simulated ALT at 55 cm and 2.8 m below the surface. This expands the soil thickness of the RIM



with 1.5 m. To allow the RIM to degrade below the elevation of the center, we additionally add excess ice to the bottom soil layer (in both coupled tiles) so that the top of RIM is only elevated only 35 cm relative to CENTER (Fig. A1), which is an approximate middle value for observed rim heights at Samoylov. The model is initiated with a soil temperature of -9 °C and fully saturated and frozen soil throughout the column. This is substantially colder that the equilibrium temperature reached by

the model. However, with total soil column of about 16 m (14 m plus 2.15-2.5 m excess ice including bottom ice, less than the observed depth of zero annual amplitude), the deep soil temperatures are spun up within the first decade of the simulation (mean increment of 0.3 °C yr$^{-1}$), after which the annual increments vary between positive and negative values.

As model forcing for the Samoylov Island simulation, we used the same model input as Westermann et al. (2016). This is based on the CRU-NCEP data for the historical period (1901-2015; Viovy, 2018). For the future part of the simulation,

this dataset uses model output from the CCSM4 climate model following the mitigation scenario RCP4.5, to calculate monthly climate anomalies for temperature, humidity, pressure and wind, and scaling factors for precipitation and radiation. These are added or multiplied to the high-frequency data from 1996-2005 from the historical (CRU-NCEP) data. This methodology follows Koven et al. (2015). The RCP4.5 scenario was chosen as it represents a strong mitigation effort, and is hence an optimistic scenario, but still show continued warming in the Arctic throughout the 21$^{st}$ century making which makes

understanding permafrost processes highly relevant.

One modification was made to this dataset in the current study, as it was observed that the model accumulated too much snow compared to observations. Detailed measurements of snow accumulation from 8 low-centered polygons from 2008 showed average snow depths of 17 cm on the rims, and 46 cm in the centers, with a total average SWE of 65 mm (Boike et al., 2013). In order to simulate similar SWE and snow depths, we scaled the precipitation with a constant factor (Pscale) of 0.6

throughout the simulation. The resulting mean annual temperature and precipitation for the whole period is seen in Fig. 4a.

### 2.3.2 Peat plateaus in Suossjavri, Northern Norway

Similar to the polygonal tundra, the peatland of Suossjavri is represented with two interacting land units. In this case we represent a single, circular peat plateau with a diameter of 10 m, corresponding to the smaller features observed at the study area. This is placed in a larger (100 m x 100 m) surrounding mire, so that the effect of the coupling is mainly on the elevated

tile. The areas of both the mire and the peat plateau can be increased to represent larger features (see Sect. 4.2), and more complex geometries can be represented by applying appropriate distance and contact length parameters. As the mire does not contain permafrost, only the peat plateau tile (PPLAT) was initiated with excess ice (Fig. A1b), starting 75 cm below the surface, and with a total excess ice thickness of 75 cm distributed down to 3.75 m below ground. Both tiles were started from fully saturated conditions and 0 °C soil temperatures. The soil water was initially unfrozen in all soil layers, except for the

ones containing excess ice, where soil (pore) water was initially frozen.

Forcing data for this location was generated in a similar way as the data used at Samoylov Island. CRU-NCEP data from nearest grid point was used for the historical part, whereas anomalies for the future (starting in year 2010) were taken



from an CCSM4 simulation following RCP4.5 scenario and added/multiplied to the reference period 1996-2005. The resulting mean annual temperature and precipitation for the whole period is seen in Fig. 7a.

## 3. Results

In the following we will look at the results of the two coupled tiles compared to the uncoupled reference tile, beginning with the polygonal tundra site in Northern Siberia (section 3.1), before looking at the peat plateau location in Northern Norway (section 3.2).

### 3.1 Samoylov Island, Northern Siberia

During the course of our simulation, Samoylov island experiences a strong increase in annual mean air temperature and modest increase in precipitation (Fig. 4a). Mean air temperatures rise from approximately -14 °C in the early 20[th] century to as high as -8 °C towards the end of the 21st century with the RCP4.5 scenario, with most of the warming happening during the 21[st] century.

Both the reference and the coupled simulations show stable permafrost with ALT between 0.45 m and 0.65 m during the historical period of the simulation (until 2010). This is in good agreement with observations, showing mean ALT close to 0.5 m (Boike et al., 2013; 2018). For snow depth and near surface soil moisture conditions, the coupled simulations show clear differences from REF (Fig. 4 b and c), and more closely mimics the observed conditions (see Boike et al., 2013; 2018 and Nitzbon et al., 2018). The simulated maximum snow depths in 2008 compares quite well with observations for both RIM (0.23 m compared to 0.16 m), and centers (0.39 m compared to 0.46) although the observations show considerable spread (see Nitzbon et al., 2018). Dry near-surface soil conditions in the RIM tile, and mostly saturated CENTER is also what is observed in this landscape (see Chadburn et al., 2017 and Nitzbon et al. 2018), a distinction which cannot be represented in the REF simulation.. With the rising air temperatures in the 21[st] century, the difference between the coupled tiles and REF become is also seen in that ALT deepens and subsidence begins in REF. Around 2030 the subsidence in REF has reached 35 cm, and by the end of the century the subsidence is more than 1 m, with the ALT still growing. In the coupled simulation, RIM remains relatively stable and elevated above the center until around 2070, almost four decades later than REF. Towards the end of the simulation RIM appears to stabilize with a subsidence of 80cm and an ALT of less than 1 m, which is also in contrast to the uncoupled REF.

CENTER experiences ALT deepening in the 21[st] century, which reaches a maximum around 2070. This follows the rapid increase in forcing temperature, and lasts until RIM has subsided below CENTER. After this point the elevated RIM tile has turned into a trough, and the top layers in CENTER starts to drain, resulting in shallower ALT. This marks the transition from a low centered to a high centered polygon.

*Soil temperatures:* The annual cycle of the soil temperature is shown in Fig. 5. In current climate (left column), the elevated rim shows annual temperature variations in the active layer of more than 20 °C, in agreement with observations (Boike





et al., 2018). At depth, the soil temperatures are higher than observed, with about -3 °C in REF and -5 °C in the coupled simulations, compared to -8.6 °C at 10.7 m depth observed during the second half of this decade (Boike et al., 2013). Here it is worth noticing, however, that these temperatures are rising, and have increased more 1 °C during the last decade (Boike et al., 2018).

5       Again, clear differences can be seen between REF and the coupled tile system. In the current climate (left column), REF and CENTER shows a very similar annual cycle, whereas the amplitude of the temperature cycle is much larger in RIM. In the active layer, the difference is almost exclusively observed during winter, when the effect of shallower snow depth is decreasing the winter insulation in RIM. Deeper into the soil column the two coupled tiles (RIM and CENTER) show much more similar temperatures, as heat exchange between the two tiles becomes more important. In the deep soil (Fig. 5c) the

temperature is therefore similar (within 0.5 °C), but around two degrees colder than in REF. Similar results are seen for the end of the century (Fig. 5b), except with opposite characteristics for the two coupled tiles. The now elevated, dry CENTER with low snow accumulation has become the tile with the large annual amplitude in the top soil, whereas RIM largely follows REF. Deeper into the soil, we again see both the two coupled tiles being colder than REF, although the difference is smaller than in the beginning of the century (Fig. 5d). Comparing the temperatures at 2 m depth from the surface in REF and the area-

weighed mean of the coupled tiles (here mean of RIM and CENTER), we find the coupled simulation on average 2.1 °C colder than REF during the 20[th] century. This difference decreases to almost zero during the transition from LCP to HCP, before the coupled simulation becomes colder 1.4 °C during the final two decades of the simulation.

      *Summer surface energy fluxes:* A clear difference between the tiles is also seen in the summer surface energy fluxes (Fig. 6). As expected, the dry RIM shows larger sensible (HFX) and lower latent (LH) heat fluxes than REF before degradation,

whereas the wet CENTER shows the opposite (Fig. 6 a). This is reversed at the degraded state when CENTER is dry and the trough (subsided RIM tile) is wet. Interestingly, the landscape aggregated values (here the mean of RIM and CENTER) is only a few W m$^{-2}$ different from the reference for these two fluxes both before and after degradation (Fig. 6a, b). We note, however, that this depends on drainage conditions. Here only surface water (infiltration excess) is removed as runoff, whereas advanced degradation in this kind of polygons is often associated with drainage also of the troughs (Liljedahl et al., 2016). This effect is

not included here, but is simulated and discussed by Nitzbon et al. (2018), and would likely move the coupled tiles towards higher Bowen rations at the degraded stage compared to both REF and the non-degraded stage. It is also likely that the difference between the reference simulation and the aggregated values would be larger with a different areal fraction of RIM compared to CENTER.

      The ground heat flux (GRDFLX) is lower during both time periods for the mean of the coupled tiles than the REF,

due to a substantially reduced flux in the dry, elevated tile (first RIM, then CENTER). This points to the effect of dry peat insulating the soil, and suggests that the lower temperatures in the coupled system could be a result of both increased summer insulation as well as the reduced winter insulation mentioned above.





Qualitatively our simulation captures the observed difference between the RIM and CENTER reported by Langer et al. (2011b), although the simulation seems sifted towards higher sensible heat fluxes and lower latent heat fluxes. This again might be related to too low water holding capacity in our simulations, as well as the lack of surface water on top of the low-centered polygon.

### 3.2 Suossjavri, Northern Norway

Figure 7 shows the soil moisture, surface elevation, ALT and snow depth at the mire location in the sporadic permafrost zone in Northern Norway. Here REF is unable to maintain permafrost, and the excess ice is rapidly disappearing over the first 3-4 decades of the simulation. After this point REF acts as a mostly saturated wetland with maximum snow depths around 1 m. The corresponding tile in the coupled system (PPLAT) experiences low maximum snow depths and dry surface conditions (Fig. 7c), which results in a stable peat plateau throughout the 20th century. Compared to observations at this location (see Sec. 2.1.2), the PPLAT shows somewhat larger ALT (0.75 - 0.9 m) in the historical period. The initial excess ice does not start to melt until around 2030, which is when both air temperatures and precipitation starts increasing rapidly. Accelerating ALT deepening is seen after 2050. At this point the mean air temperature has stabilized at about -1 °C and precipitation around 650 mm. However, the acceleration of the ALT deepening appears to be driven by feedbacks in the system. First, we have the melt-subsidence-snow feedback. As the ATL deepens and excess ice melts, the peat plateau subsides, leading to more snow remaining in this tile and smaller heat loss during winter, which again enhances summer melt. Next, the subsidence also results in a thinner layer of dry peat as the water table is largely controlled by the elevation relative to the mire, which lowers the insulation during summer. Combined with the direct effect of water from the excess ice melt increasing the soil moisture in PPLAT, this leads to a melt-subsidence-soil moisture feedback. The surrounding MIRE is largely unaffected of the presence and disappearance of the elevated peat plateau as it is here simulated to be about two orders of magnitude larger. Hence, REF and MIRE develops very similarly after the initial excess ice in REF has melted.

*Soil temperatures:* The soil temperatures in the coupled tiles differs substantially in the present, non-degraded state (Fig. 8 a, c). Whereas REF and MIRE have nearly identical annual temperature cycles near the surface, PPLAT deviates on several points. First of all, the elevated PPLAT shows cold winter soil temperatures (as low as -7.6 °C in January), compared to a constant, zero-degree temperature in MIRE and REF. Furthermore, PPLAT responds quicker to the onset of both summer and winter, with both MIRE and REF shifted somewhat to warmer temperatures in late summer and colder during spring. One key factor controlling these differences is the low snow accumulation in PPLAT, which lead to both increased annual temperature cycle near the surface, and an earlier onset of spring due to less energy going to snow melt. Another factor is the higher soil moisture in REF and MIRE (both mostly saturated), which due to the high heat capacity of water will delay the soil response to changing atmospheric temperatures.

Below the depth of zero annual amplitude, PPLAT sees warm permafrost conditions at zero °C, whereas the MIRE and REF is close to 3 °C. Here there is a slight difference between the REF and MIRE, with the former being about a quarter of a degree colder, due to the memory at this depth from ice melting earlier in the simulation (Fig. 7).



After degradation (Fig. 8 b, d), the difference near the surface is marginal between all three realization. At this point there is no elevation difference, and hence no difference in snow accumulation or other forcing at the surface. The difference is then confined to the lower soil layers, where the PPLAT tile is still warming after the ice melt

*Summer surface energy fluxes:* The different snow and soil conditions between the MIRE and PPLAT are also clearly visible in the summer surface energy fluxes (Fig. 9). In the present, undegraded state, the PPLAT tile shows almost opposite HFX and LH fluxes compared to both MIRE and REF, which again are practically identical. Whereas the MIRE and REF both shows three to four times larger LH than HFX, the opposite is the case for the dry, elevated PPLAT. At this location, unlike the polygonal tundra site, the average over the two coupled tiles would differ substantially from REF. As the MIRE is two orders of magnitude larger than PPLAT in the present setup, the aggregated fluxes is only be marginally different than what a single MIRE tile (similar to REF). However, observed peat plateaus can occupy a large area of the landscape (as also seen from Fig. 1b), and representing MIRE and PPLAT of more equal size will result in larger differences in the aggregated fluxes.

For the ground heat flux (GRDFLX) the differences are smaller, but still substantial. The elevated PPLAT receives on average less energy from the surface during summer compared to both RIM and MIRE. With colder temperatures at depth in this tile, this points to the insulating effect of dry peat as being a contributor to sustaining permafrost, in addition to the above-mentioned winter effect from shallower snow depths. In the degraded phase, the difference between all three realizations have nearly vanished, as the PPLAT is no longer elevated from the MIRE. Here only a slightly larger GRDFLX (scarcely visible) shows that temperatures below are still adjusting from after the ice melted.

## 4. Sensitivity studies

To further investigate the importance of the different processes in the coupled system, we perform two sets of sensitivity studies. First, we look at the effect of turning on an off the different lateral fluxes at both locations, before looking further at the effect of the distance parameter (D) for the simulation of the peat plateau location.

### 4.1 Snow, water and heat coupling

Figure 10 shows the surface elevation in the elevated tile (RIM/PPLAT) at both locations for different combinations of lateral fluxes. Here the tick blue and red lines represents the reference (similar to REF) and the fully coupled (similar to RIM/PPLAT) realizations in section 3, respectively.

For the polygonal tundra site (Fig.10a) the snow effect alone (thin red) gives similar results as the fully coupled simulation during most of the time period. The difference is clear only towards the end, when the snow-only experiment continues to melt and subside with a trough approaching 1 m depth (corresponding to 1.35 m subsidence), whereas the fully coupled system stabilizes with a 45 cm trough. Individually adding lateral water (yellow) or heat (purple) fluxes has opposite effects, decreasing and increasing the melt process, respectively. The snow + water coupling (green) results in an almost stable rim throughout the simulation, subsiding only about 10 cm before the end of the 21st century, whereas the snow + heat coupling



(thin blue) results in about 10 years earlier subsidence than the fully coupled realization, but eventually stabilizing at almost the same depth.

At the location in Northern Norway, the combined effect of snow and water coupling is needed to simulate a stable peat plateau throughout the 20[th] century (Fig 10b). Only the fully coupled (tick red) and the snow+water coupling (green) can
represent stable permafrost, whereas all other simulations see degradation starting within the first decades of the 20[th] century and ground ice disappearing entirely before 1970. Adding the lateral heat flux to the reference setup (purple) has little effect. However, in combination with the snow and water coupling, the heat flux is speeding up the melt so that the peat plateau disappears two decades earlier than without the heat coupling.

Seen together, it appears that all three lateral fluxes are important at both locations. Compared only to the reference
simulation, the effect of snow redistribution is largest, followed by the effect of water coupling, whereas the effect of the lateral heat flux alone is marginal. However, both snow and water coupling act to cool the elevated tile compared to the CENTER/MIRE, as seen by the delayed subsidence. Hence an increased thermal gradient is produced that increases the effect of the lateral heat flux. The result is that the stabilizing effect of snow and water fluxes are reduced, and degradation speeded up. The relative effect of the different processes is therefore complex, and must be seen in combination with the other fluxes.

The influence of the different lateral fluxes is to some degree sensitive to the key parameters and how it is implemented. This is especially the case for snow redistribution, which in our simulations was found to be the most important coupled process. Here we redistribute all solid precipitation from the tile with the highest surface elevation (soil + snow), once a minimum snow depth is reached ($Hs_{min}$). Increasing (decreasing) this limit will decrease (increase) the effect of snow redistribution in the simulation. Similarly, the thermal and hydraulic conductivity of the soil will determine the effect of the
heat and water fluxes, respectively. However, the effect of lateral heat flux was only important in combination with snow and/or water coupling, as there must already be a thermal gradient between the tiles before it can have an effect. Finally, the lateral heat and water fluxes will depend on the geometry of the system, in particular the distance parameter (D).

## 4.2 Distance parameter (D)

To test how sensitive the system is to the distance parameter (D), we perform another sensitivity test for the mire location. As
seen from Eq. (5) and (6), both lateral water and heat fluxes depend linearly in this parameter. However, the water has the potential for draining fast, and becoming in equilibrium while the heat conduction is generally much slower and remove temperature differences less efficiently. To test a wide range of parameter values we simulate a larger system than in section 3. Again, we simulate a circular elevated tile, but scale both the elevated PPLAT and the surrounding MIRE by a factor of 100 in each horizontal direction, and test length scales from 0.2 m to 500 m.

Figure 11 shows the resulting surface elevation in the peat plateau (a), as well as the lateral heat (b) and water fluxes (c) shown as 10-year running averages. Here we see that for the most part larger distances correspond to earlier melt and subsidence. This is clearly a water effect, as the simulated annual horizontal heat flux (HHF) is small, and scales almost linearly with D[-1] and snow redistribution does not depend on this parameter. Hence the main mechanism appears to be that larger D



gives slower lateral water flux, and hence larger soil thermal conductivity at PPLAT. Only with very small or large D is this picture reversed. Going from distance parameter of 0.2 m to 0.5 m gives instead a slight delay in degradation. For such small values of D, the changes in lateral heat fluxes are important, whereas lateral water fluxes are almost instantaneously and does not change noticeably. Hence the effect is that larger D gives slower degradation. In the other end of the simulated range,

degradation is also delayed when going from 50 m to 100 m, and further to 500 m. With such large D values, the drainage is much slower, and full drainage on the annual time scale is no longer realized. Hence other processes like increased heat capacity from increased soil moisture might be more important than the conductivity effect of reduced drainage.

While this does not translate directly to the effect of changing size of the PPLAT tile, this sensitivity analysis shows what the two lateral subsurface processes are important on different scales, showing that the results from the previous section

depends strongly on the geometries and sizes of the structures.

## 5. Discussion

With a relatively simple two-tile system, we have been able to simulate observed microtopographic changes associated with ice-rich permafrost degradation. As a direct effect, we have seen that this altered mean soil temperatures, active layer thicknesses, timing of permafrost degradation, soil moisture conditions and surface energy balance fluxes. In the following,

we will discuss limitations and sources of errors in the current study (5.1), how this method might be implemented in large-scale models (5.2), and possible implications for simulations of the PCF (5.3).

### 5.1 Limitations and sources of errors

The method applied here is by design a minimal approach to include the important lateral processes in permafrost landscapes, where the number of new parameters (see Fig. 3) have been kept at a minimum. As capturing the detailed properties at the two

specific test sites have not been objective of this study, the different parameters have not been fine-tuned, neither for the default NoahMP model or the new tile geometry parameters. As noted above, there are differences between the observed properties at the two locations, and what is simulated here. In particular, the simulations showed considerably warmer permafrost temperatures and larger Bowen ratios at Samoylov Island, whereas the peat plateau at Suossjavri appear more stable in the simulations than what is observed. In the following we will discuss some aspects of the two-tile system that were found to be

particularly important for our simulations, as well as other properties and processes ignored here, which might explain these discrepancies.

*Snow*: First, the minimum snow depth was found to be a key parameter. As seen in Fig. 10, the timing of the degradation at both locations were sensitive to the snow redistribution. This is in agreement with previous studies on the effect of snow on thermal regime (e.g. Ginsås et al., 2016) For our simulations, it was found that a higher minimum snow

accumulation limit was needed for the peat plateau (10 cm) than for the polygonal site (5 cm), in order to simulate stable conditions in the beginning of the simulation and degradation within the current century. We note that the end-of-winter snow



depths at both locations are within the observed range. However, this parameter should in the future be studied further, and ideally be linked to simulated properties like vegetation height.

*Excess ice initialization*: Another key aspect of the coupled system is how the excess ice is initiated. The depth at which the excess ice was inserted in the soil column (Zextop) could in theory be set to the observed ALT. Test simulations
revealed, however, that inserting this at a too large or too shallow depth would result in either a too stable or too unstable system, respectively. Inserting this at the depth of the simulated ALT was therefore found to be important to simulate reasonable degradations. Likewise, the density of the excess ground ice is important for how fast the system evolves once degradation starts, and at the polygonal tundra site, what is the new stable state. The values used here is to some extent based on observations and expert judgement, but still a certain degree of trial and error was needed, in particular to simulate a stable
trough at tundra site, without a continued, unrealistic deepening of the RIM tile.

*Soil properties*: An important limitation in the current model system is the uniform soil properties, both with depth and between the tiles. This is a simplification inherent in the current NoahMP model, which does not consider different soil types at different depths, which included also our implementation of organic fractions. This meant that in order to represent observed soil properties near the surface, in particular the high porosity typical for peat, we likely also simulated too large
porosities deeper into the soil. The effect of this is that too much soil water is changing phase in the active layer every year, dampening the temperature signal from the surface. Another effect might be that the energy needed to thaw the soil might be exaggerated, as the initial amount of soil (pore) ice is too large. Also related to soil properties, is the lack of new excess ice formation and of explicit representation of surface water above the soil column, both of which might act to slow down degradation. While the former is a more uncertain process for which appropriate model formulations are lacking, the latter has
been accounted for in the companion paper (Nitzbon et al., 2018).

*Larger scale hydrology:* As shown in the companion paper (Nitzbon et al., 2018), the surrounding hydrology is important for the stability of permafrost in polygonal tundra. Here we have simulated one hydrological setting, where surface water is removed as runoff, but otherwise the polygon is detached from the surroundings. Observations from Samoylov reveals, however, that very different hydrological conditions can be found even on this relatively small island. Simulating instead
surface water in low-centered polygons, or waterfilled troughs in the degraded, high-centered stage, would modify the results presented here.

*Vegetation:* Another factor that could modify the results is the influence of vegetation, which has not been considered here. In particular, the appearance of new vegetation troughs might lead to an increased insulation, and act as a negative feedback to the degradation of the polygons, interacting also with the local hydrology.
*Other lateral processes:* Finally, we note that there are other lateral processes that we have not accounted for, including lateral erosion and heat advection associated with lateral water flow. While we have included the effect of lateral heat flux on the average temperature, this heat flux will in reality often lead to permafrost thaw and erosion near the margins. Furthermore, our simulation only includes lateral water flux near the surface, whereas a number of studies have demonstrated



that deeper water movement and related heat advection might affect soil thermal conditions (Kurylyk et al., 2016; Sjöberg et al., 2016). Both of these processes likely contribute to the degradation of peat plateaus currently observed in Northern Norway.

Despite these limitations, our results show a major improvement in terms of representing current permafrost conditions at the two locations, and will help improve permafrost processes in LSMs particularly on lateral transport. The
discrepancies with current observations are consistently smaller for the coupled simulation compared to REF. Considering also that observations from both locations show considerable spatial variation, and current temperatures at the polygonal site is rapidly increasing, we consider these differences acceptable, mainly influencing the timing of the future permafrost degradation. Furthermore, some of the biases seen here could be mitigated with vertically varying organic fractions and types, as well as dynamical vegetation, which is already included in several large-scale LSMs.

**5.2 Interactive tiling in coupled ESM simulations?**

Having demonstrated and discussed the large impact lateral fluxes may have on the simulated permafrost state and evolution we turn to the possibilities for representing these processes in global scale ESMs. Here, we have used NoahMP as a test model, in which subsurface processes are represented comparable to LSMs used for global climate simulations. In the companion study, Nitzbon et al. (2018) have demonstrated that the same basic method can be utilized in a completely different model,
which suggests that the method is model independent. Implementation in a large-scale LSM, also with full biogeochemistry, should therefore be more of a technical rather than conceptual challenge, as is applying the modified model online in an ESM framework. From the theoretical side, the challenge is mainly the scale gap between the small-scale units considered here (on the order of 10 – 100 m) to the 100-km scale typical for current global simulations. However, mentioned in section 2.3, the concept of self-similarity offers the possibility to represent larger landscapes with a small set of coupled units.

As the simplest implementation, we suggest using the two-tile structure outline here to represent a fraction of a grid-cell in a large-scale model, alongside the default LSM simulation. Using for instance the maps from Olefeldt et al. (2016) to identify areas of ice-rich permafrost susceptible to thermokarst development, these can be represented by a separate land unit consisting of two laterally coupled tiles. In the simplest form, this would require only some representative geometries for each type of permafrost landscape. Even with relatively crude estimates of representative geometries in each permafrost region, this
would for the first time allow including the effects of lateral heat, moisture and snow fluxes on permafrost degradation in global models. As suggested by field observations (Liljedahl et al., 2016), these can have substantial effects on the evolution of the permafrost region which cannot be represented when assuming homogeneous permafrost and ice distributions.

The method described here is, however, not limited to a two-tile structure. As demonstrated by Nitzbon et al. (2018), the same basic formulation can be applied also with three tiles, and with water exchange with an external reservoir. In such a
configuration, the coupling method already gives a substantially higher level of complexity and realism for the specific site studied (Samoylov Island), although the number of input parameters is correspondingly increased. From a system with three coupled tiles, one can expand the method further to more tiles representing physical locations in a single system (like surrounding hills or waterbodies), or to an ensemble of two- or three-tile systems with different geometries within a single grid



cell. However, with the computational cost of current LSMs used in ESMs, this would relatively quickly become a large computational burden.

**5.3 Interactive tiling to improve simulations of future permafrost-carbon-feedback?**

Improved representation of the PCF is a key motivation for the model developments outlined here, and we will therefore in the following discuss qualitatively how it might be affected by the geophysical changes seen here.

As a first order effect, the PCF depends on how much, and when, currently frozen ground is thawed and soil carbon exposed to microbial decomposition. As substantial changes in permafrost state (e.g. Fig. 5) and timing of degradation (Fig. 4 and 7) have been demonstrated here, it is clear that also the simulated PCF will be affected by implementing this method in a full ESM. The CMIP5 ensemble of climate models showed drastically different permafrost area (Koven et al., 2013a), which have been a major contribution to the uncertainty in the PCF. More recent simulations with improved ESMs (McGuire et al., 2018) still shows huge differences in simulated PCF and vegetation response in the permafrost region. This is despite the fact that they all still lack the kind of sub-grid processes included here. Our results therefore suggest that current large-scale LSMs used in climate models might also have large biases in the amount of permafrost and when it could thaw due to the lack of lateral processes discussed here (e.g. Rowland et al., 2015).

On top of this, our method shows non-linear behavior and thresholds that are not found in the reference simulation. Both the polygonal tundra and peat plateau settings show more stable conditions when lateral fluxes are included, before relatively rapid changes were initiated. One reason for this, is the non-linear effect snow has on the insulation of the soil during winter. When considering a grid-cell mean snow depth, the effect of slightly increased or decreased snow accumulation is often small. However, in a coupled system where snow redistribution depends on dynamical micro-topography, even a small increase in snow accumulation can be enough to initiate a change. Once elevated features begin to degrade, they cannot be reversed, and may not even stabilize before reaching a new state, even if the change in forcing were temporary. This kind of behavior has implications for so-called overshoot scenarios, which is being discussed in relation to the more ambitious mitigation scenarios (Comyn-Platt et al., 2018).

Accurate simulation of the PFC depends also on the soil moisture conditions under which permafrost thaws and carbon is mobilized. This has been demonstrated both in lab measurements (Elberling et al., 2013; Schädel et al., 2016; Knoblauch et al., 2018) and model simulations (Lawrence et al., 2015), showing that realistic simulation of local hydrology is important for PFC simulation. Our results here capture key soil moisture conditions observed in the two kinds of landscapes simulated, which cannot be represented in the reference simulation. Furthermore, our method simulates the rapid transitions in soil moisture conditions that are associated with thermokarst development, which might in itself be an important factor for the amount and partitioning ($CO_2$ vs $CH_4$) of carbon release to the atmosphere. Incubation measurements have shown that the largest greenhouse gas production from thawing permafrost could be realized when previously wet soil was decomposed under drained conditions (Elberling et al., 2013). This is exactly the process observed in the centers when LCP transition into HCP, which again demonstrates that the processes included here might play an important role in accurately simulating PFC.




Finally, the sub-grid tiling proposed here will likely impact dynamical vegetation modeling in the permafrost region, which is important for understanding the future carbon signal from this region. Both greening and browning of the Arctic have been observed over the recent past and both could be expected also in the future, although a general greening and increased carbon storage in vegetation is what models are predicting (McGuire et al., 2018). The processes related to PFT migration and
competition might, however, be different when the sub-grid variability in snow and soil conditions represented here are included. This further exemplifies how the method applied here might offer new details and added realism when simulating land-atmosphere interactions in the Arctic permafrost region.

## 6. Conclusions

Here we have simulated dynamically changing microtopography and related lateral fluxes of snow, water and heat with a two-
tile model approach, and applied it to two very different kinds of permafrost landscapes. The main findings of our investigation are as follows:

1.  Currently observed degradation processes in both polygonal tundra and peat plateaus could be simulated with a simple tiling approach accounting for changing micro-topography and small-scale lateral fluxes. This included representing the transition from low- to high-centered polygons and the transition from a stable to degrading peat plateau in the
sporadic permafrost zone.
2.  The timing and speed of degradation at the polygonal site differed strongly between the two simulations. Whereas reference simulation showed slow, but continuing degradation which did not stabilize during the simulation, the coupled tiles showed a delayed onset of permafrost degradation, followed by a more rapid ice melt and subsidence, before the system stabilized in a new state with an elevated, dry center and wet trough.
3.  Deep soil temperatures differed substantially between the reference simulation and the two-tile system. For the polygonal tundra site at Samolylov Island, the simulated temperatures at approximately 13 m depth was about 2 ºC colder with coupled tiles in the current climate.
4.  Dry near-surface soil conditions in the elevated features could be represented at both locations, with substantial effect on surface energy balance fluxes.
5.  Sensitivity studies showed that lateral fluxes of snow, water and heat all affected the stability of the permafrost at both locations, with their relative contribution depending on the distance parameter.

These results show that lateral fluxes and changing microtopography have a strong impact on simulated permafrost extent and conditions. Together with the companion study by Nitzbon et al. (2018), we have demonstrated that this can be realized to a first order with a simple and computationally effective tiling approach. Applying the proposed method in land surface
models with full biogeochemistry shows significant potential to drive simulations of the permafrost-carbon-feedback towards reality.





*Code availability.* The NoahMP model code is available at https://github.com/NCAR/hrldas-release. Modifications to the code implemented here are available from the corresponding author upon request.





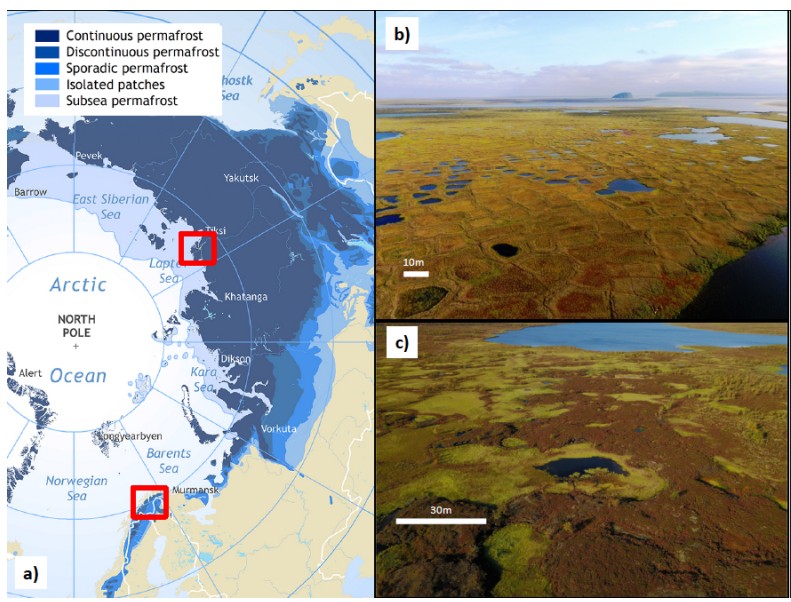

5  **Figure 1: a) Location of the two test sites on top of the map of permafrost distribution in the Arctic (Brown et al. 1998). b) example of low-centered polygons on Samoylov Island, Northern Siberia (Photo: Sebastian Zubrzycki). c) Example of peat plateau near Suossjavri, Northern Norway (Photo: Sebastian Westermann). The two sites are located in the continuous and sporadic permafrost zones, respectively.**





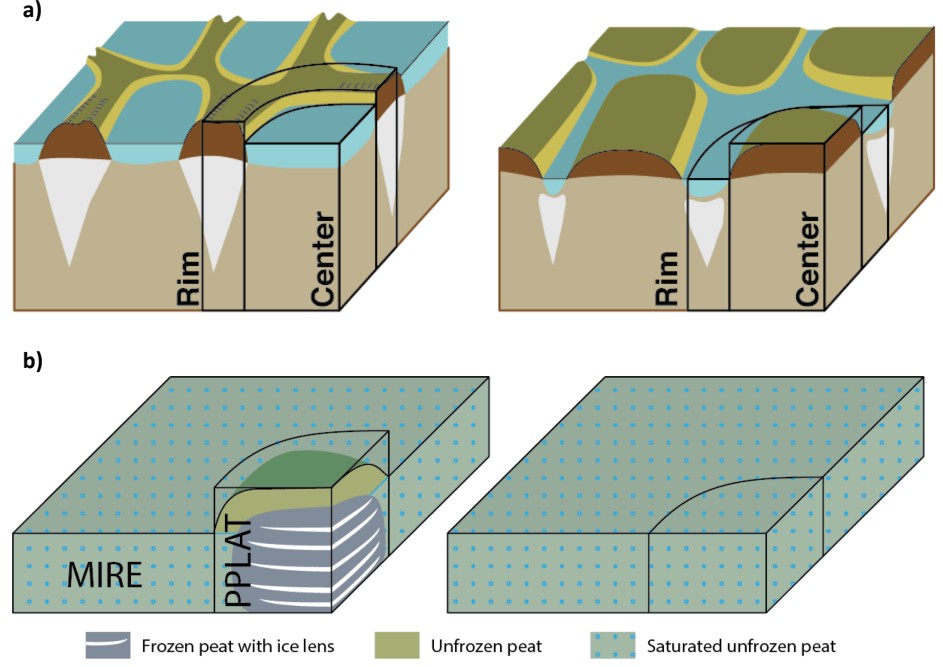

Figure 2: a) Schematic presentation of undegraded, low-centred polygon (left) and degraded, high-centred polygons (right) with corresponding tiles used in this study (adapted from Liljedahl et al. 2016). b) Schematic presentation of peat plateau in a surrounding mire (left), and the mire after the peat plateau has degraded (right), with corresponding tiles.





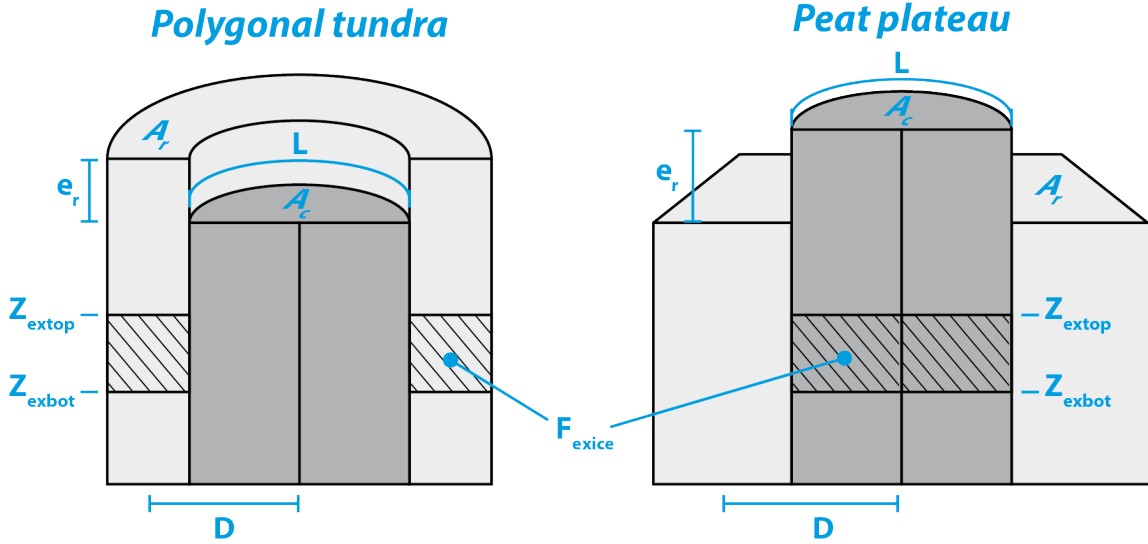

**Figure 3: Schematic presentation of the two-tile system with geometry parameters. Parameter values used for the two locations are listed in Table 1.**





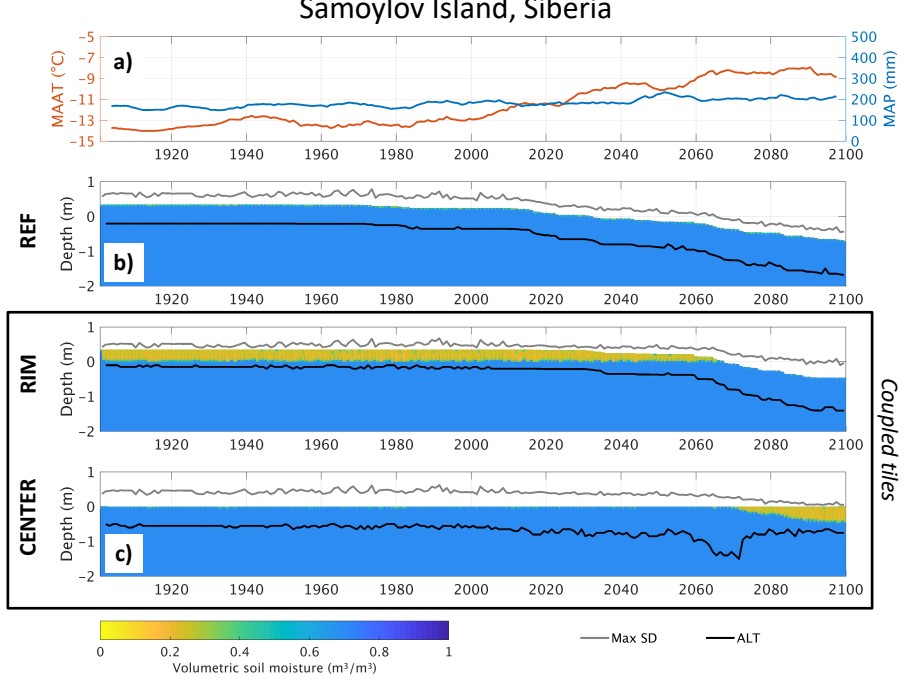

**Figure 4: a) 10-year running average of mean annual air temperature (MAAT) and mean annual precipitation (MAP) at Samoylov Island. Soil moisture and surface elevation are shown as colored region in b) reference simulation, c) and in the coupled tiles. Note that both the surface elevation (relative to the CENTER tile) and the unsaturated soil (orange and green colors) change in the coupled tiles as excess ice melts and the lateral fluxes changes. Maximum annual snow depth (MaxSD) and active layer thickness (ALT) are shown as gray and black lines, respectively.**



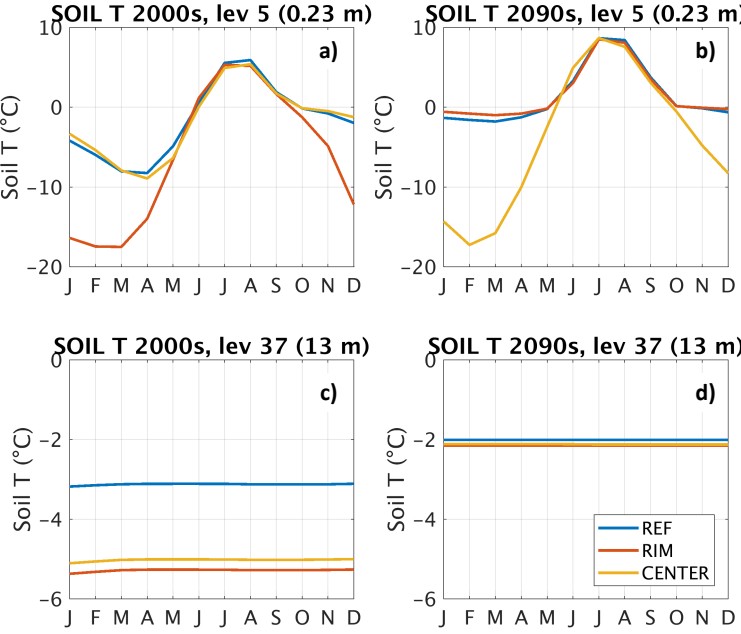

**Figure 5: Average annual temperature cycle during first (a, c) and last (b, d) decade of 21st century at Samoylov Island, in the 5th model layer (a, b; 0.23 m below surface) and the 37th model layer (c, d; approximately 13 m below reference height). See layer depths in Fig. A1a.**





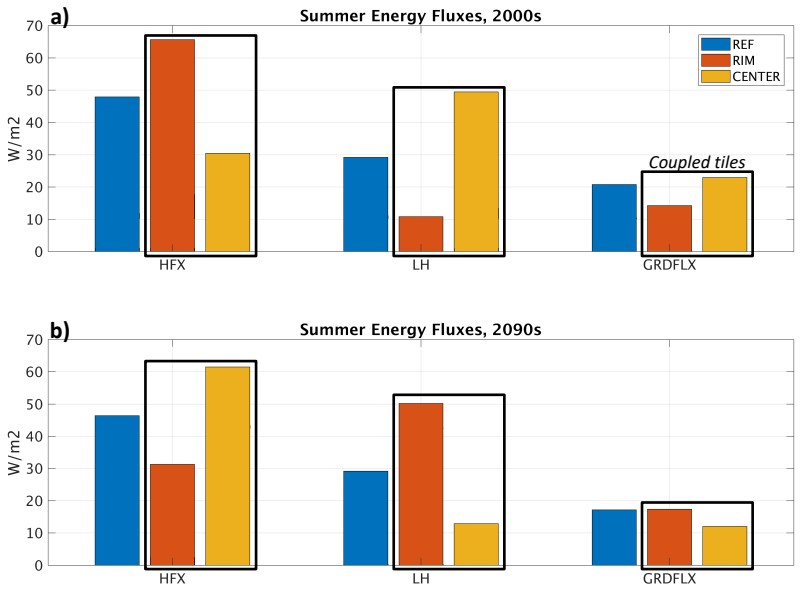

**Figure 6: Summer surface energy fluxes at Samoylov Island during first (a) and last (b) decade of the 21st century in reference simulation (REF) and coupled tiles (RIM and CENTER). HFX: sensible heat flux, LH: latent heat flux, GRDFLX: ground heat flux.**



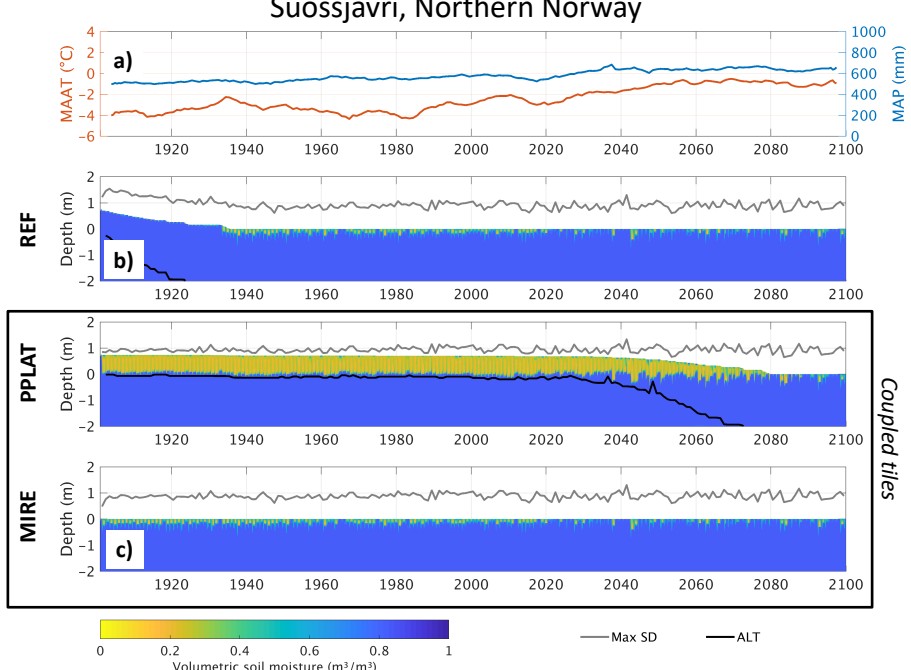

**Figure 7: a)** 10-year running average of mean annual air temperature (MAAT) and mean annual precipitation (MAP) at Suossjavri, Northern Norway. Soil moisture and surface elevation are shown as colored region in **b)** reference simulation, **c)** and in the coupled tiles. Note that both the surface elevation (relative to the CENTER tile) and the unsaturated soil (orange and green colors) change in the coupled tiles as excess ice melts and the lateral fluxes changes. Maximum annual snow depth (MaxSD) and active layer thickness (ALT) are shown as gray and black lines, respectively.





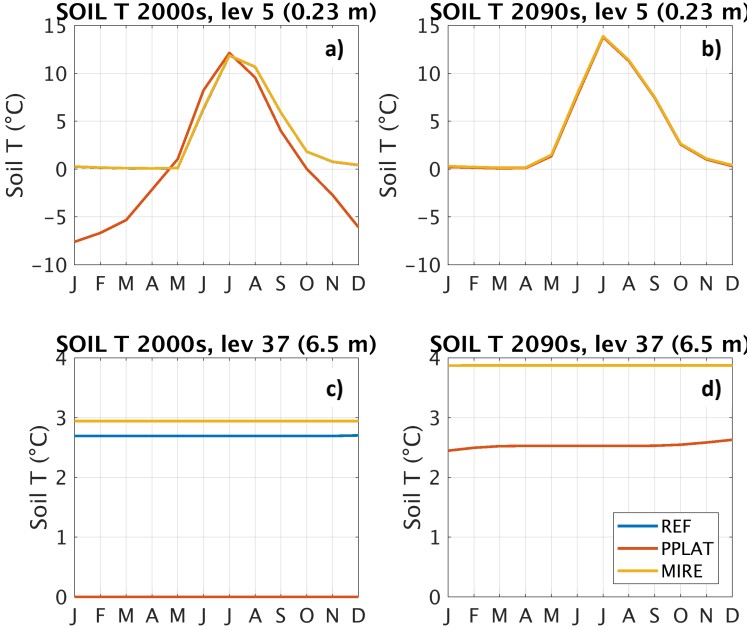

**Figure 8: Average annual temperature cycle during first (a, c) and last (b, d) decade of 21st century at Suissjavri, Northern Norway, in the 5th model layer (a, b; 0.23 m below surface) and the 37th model layer (c, d; 6.5 m below reference height). See layer depths in Fig. A1b.**



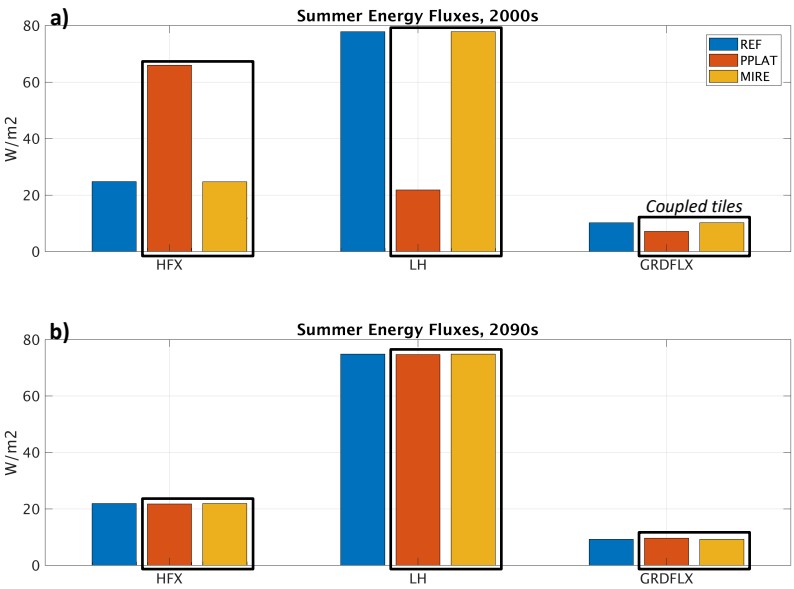

**Figure 9: Summer surface energy fluxes at Suossjavri, Northern Norway, during first (a) and last (b) decade of the 21$^{st}$ century in reference simulation (REF) and coupled tiles (RIM and CENTER). HFX: sensible heat flux, LH: latent heat flux, GRDFLX: ground heat flux.**



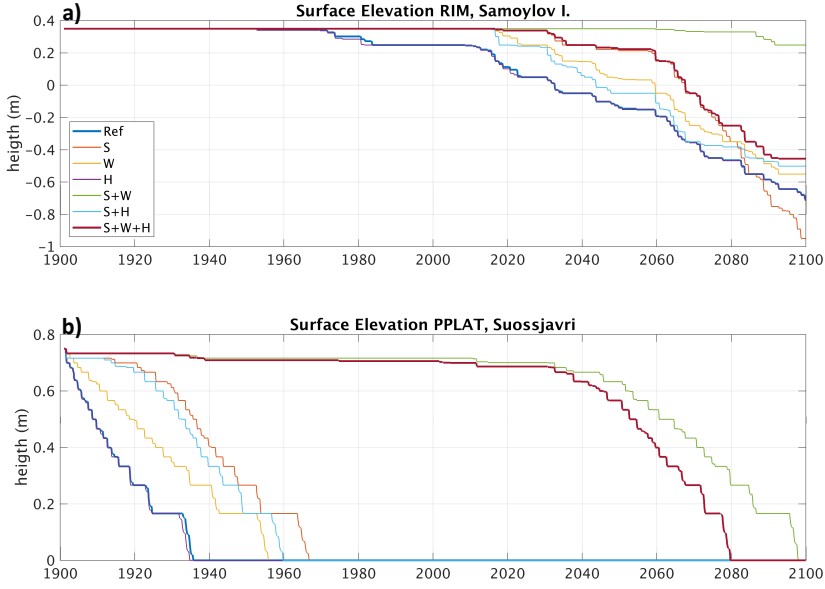

**Figure 10: Surface elevation of RIM relative to CENTER at Samoylov Island, Siberia (a), and of PPLAT relative to MIRE at Suossjavri, Northern Norway (b) for different combinations of lateral fluxes. Thick blue line represents reference simulations (REF; no lateral fluxes), and thick red line represents fully coupled simulation. S: snow, W: water, H: heat.**



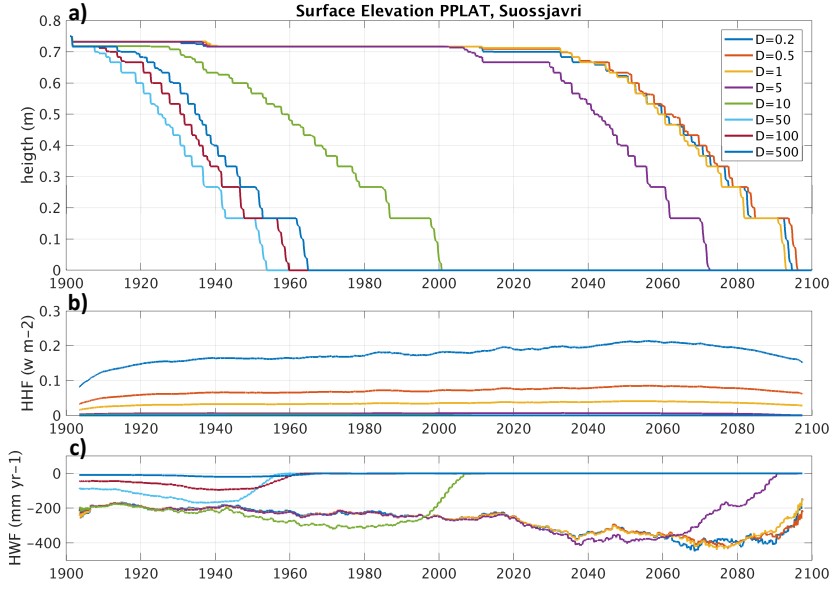

**Figure 11: Surface elevation of PPLAT at Suossjavri, Northern Norway (a) for different values of distance parameter (D) between the two interacting tiles, and corresponding horizontal heat flux (HHF; b), and horizontal water flux (HWF; c). Note that the area of PPLAT is different (larger) here than in the standard simulations (Fig. 7), giving a different evolution for the same (D=10) distance parameter.**



| Location: | Ar(m2) | Ac(m2) | L(m) | D(m) | e(m) | Zextop(m) | Zexbot(m) | Fexice (%) |
|---|---|---|---|---|---|---|---|---|
| Sam | 39.3 | 39.3 | 22.2 | 4.27 | 0.35 | 0.55 | 2.8 | 66.7 |
| Suo | 9.92e3 | 78.5 | 31.4 | 10.0 | 0.75 | 0.75 | 3.75 | 25.0 |

**Table 1: Tile geometry and excess ice distribution. See detail in Fig 3.**

| Location: | OrgF(%) | Scenario | T0(°C) | $Hs_{min}$ (m) | Pscale | Soil type |
|---|---|---|---|---|---|---|
| Sam | 50 | RCP4.5 | -9.0 | 0.05 | 0.6 | Silt |
| Suo | 80 | RCP4.5 | 0.0 | 0.1 | 1.0 | Silt |

**Table 2: Soil properties and initial conditions.**





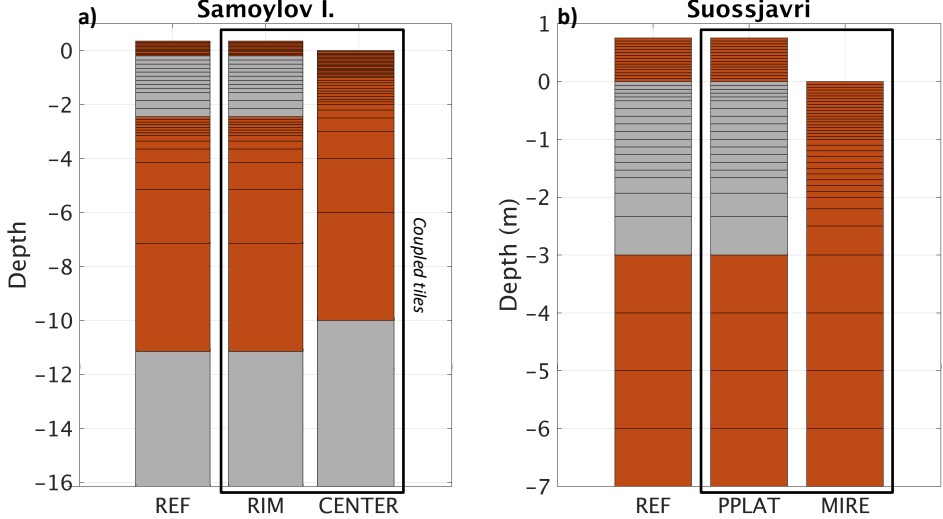

**Figure A1: Vertical resolution and at Samoylov Island (a) and Suossjavri (b). Gray colours indicate layers initialized with excess ice. Excess ice initiated in bottom layer at Samoylov Island is used to adjust the rim height ($e_r$) independently of near-surface excess ice.**



*Author contributions*. KSA and SW designed the study. KSA implemented the code, carried out the simulations and wrote the manuscript. LM made Fig. 2 and Fig 3. All authors interpreted the results and contributed to the manuscript. SW and HL secured the project funding.

*Competing interests*. JB and ML are members of the editorial board of The Cryosphere. All other authors declare that they have no conflict of interest.

*Acknowledgements*. The authors gratefully acknowledge the support of the Research Council of Norway for the PERMANOR project (no. 255331) and FEEDBACK project (no. 250740), support from the strategic research initiative LATICE (Faculty of Mathematics and Natural Sciences, University of Oslo https://mn.uio.no/latice), as well as simulation resources provided by NOTUR (project no. NN9489K) and the Department of Geosciences, UiO.





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
