# Peer review of "Thaw processes in ice-rich permafrost landscapes represented with laterally coupled tiles in a Land Surface Model"

_The Cryosphere, 2018_

## Referee Comment (RC1) · Anonymous Referee #1 · 19 Dec 2018

This paper describes how small-scale surface heterogeneity due to excess ice can be, in a relatively simple way, implemented in land surface models, in this cas, the NOAHMP LSM. A companion paper described a similar work with the Cryogrid model. The motivation of the work is clearly and convincingly laid out, the paper is well structured, easy to read and generally well written (except for frequent systematic grammatical errors). The spirit of the paper is that this work should be seen as a proof of concept, and it is made rather clear that the implementation of such an approach in ESMs will not be an easy task.

The methods are described very clearly, and they appear to me appropriate in terms

of complexity in the sense that the proposed scheme appears to be on a similar level of complexity as the rest of the model this scheme was implemented in. One might wonder whether some effort should have been devoted to implementing excess ice formation; the possible long time scales involved in the excess ice aggradation could be an argument to discard that option, given that the type of models this approach is designed for is made for centennial-scale simulations, at best.

The discussion of the limitations of this work is honest. I would have liked to see a more thorough analysis of the sensitivity of the model to some critical parameters, in particular those linked to snow; maybe some sensitivity tests might be in order. In the discussion (sections 5.1 and 5.2), it would have been good to provide the reader with some more quantitative (if possible) estimates of the importance of neglected processes, and with corresponding priorities in future developments. Concerning the implementation in ESMs, it is clear that heterogeneity linked to excess ice is relevant only on a small part of the globe. In many other places, the most relevant heterogeneities are linked to vegetation, orography, or other factors. Can the authors think of a more general (globally relevant) tiling concept in which the tiling linked to excess ice could be integrated?

In conclusion, I definitely think that this paper should be published if the points above and some specific points mentioned below are addressed. This should only require minor modifications.

Specific points:

- P5 L28: FEXICE (and similar variable names in the text): In the figure, you use $F_{EXICE}$ (EXICE as an index), so please do so in the main text, too.

- P6 L27: In the equation and in the text, $K_{sat}$ is a constant. Call it $K_{sat,0}$ for $K_{sat}$ at the surface, to prevent confusion.

- P7 L10: "The lateral ground heat flux [...] between two grid cells with overlapping

soil depth. . ." Cells or tiles? Probably tiles.

- P7 L17-18: Not clear why the elevated tile is used as a reference. In most models, there is no excess ice yet, so it might have been more appropriate to use the lower tile as a reference (especially because you do not use stagnant water at the surface anyway).

- By the way, it would have been nice, in the discussion, to sped a few lines on discussing how taking into account stagnant water could have changes the results. In my opinion, it could have very major impacts.

- P7 L29: "This expands the soil thickness of the RIM with 1.5 m". Wouldn't "by 1.5 m" be better English?

- P8 L2: "we additionally add excess ice to the bottom soil layer (in both coupled tiles)": In the figure it looks like the 35 cm excess ice are added to the lowest layer only in the lower tile. Please clarify.

- P8 L14: "but still show continued": -> shows. In many places, there are wrong or missing s's (wrong plurals, wrong conjugation). Please go through the text carefully.

- P8 L14: "making which makes"

- P9 L2: I understand why you introduce figure 7 here before figures 5 and 6, but I think that the figure numbers should be in order of appearance in the text nevertheless.

- P9 L20: "simulation.." (only one point needed)

- Same line: "become is": ?

- P10 L19: Why do you call the sensible heat flux HFX? Doesn't make much sense to me.

- P11 L23: Replace whereas by while (I think)

- P12 L16: scarcely -> barely?

- P13 L26: "becoming in equilibrium": Are you sure that this is good English?

- P14 L3: replace instantaneously by instantaneous (and does by do on the same line)

- P14 L27: As said before, a sensitivity test showing the effect of the snow parameters would have been interesting. Or would that be too model-specific?

- P15 L23: "Simulating instead surface water in low-centered polygons, or waterfilled troughs in the degraded, high-centered stage, would modify the results presented here." As said before, I'd like to see a discussion how this would modify the results (in your expert opinion)

---

## Referee Comment (RC2) · Anonymous Referee #2 · 21 Dec 2018

In this paper, the authors take steps towards an ability to represent in a large-scale model the important lateral snow redistribution, water, and heat processes that impact the trajectory of permafrost thaw and related processes in different permafrost landscapes. The approach is parsimonious, which I like. The authors propose to represent these systems with just two 'tiles' (rim and center for polygonal tundra), rather than explicitly modeling the full complexity of the heterogeneous landscape. I like this approach as it does lend itself to potential inclusion across the pan-Arctic. A significant limitation is that the model is not explicitly modeling the formation of these permafrost landscape features. Instead, the goal is simply to be able to simulate the transition from a low-centered to a high-centered polygon. This is a reasonable first step and the

authors acknowledge this limitation. Clearly, to have 'full' confidence in the model, one would want it to be able to simulate the full set of physical processes that drive both the formation and the decay of low-centered polygons. Nonetheless, this is a practical first step that is clearly an improvement over the current 1-tile assumption that cannot at all account for the real spatial heterogeneity of the system.

Overall, I enjoyed reading this paper and I find it suitable for publication with a few relatively minor revisions and clarifications.

Specific comments

1. When the Noah-MP model is introduced, it would be good to explain why Noah-MP is being used instead of any other model. I believe that it is because of the lateral flow capabilities in WRF-hydro, but that capability isn't introduced until section 2.2.4.

2. P. 8, line 2 typo: "only elevated only"

3. I wonder if the "coupled" is the best way to reference the multiple tile simulations. Coupled can mean a lot of things in different contexts. Perhaps you could rename as Reference and Tiled or Single column and Two column or something else that is more descriptive.

4. Figure 5: Why is the ref simulation at depth so much warmer than either the RIM or CENTER simulation?

5. P.9, Line 16: "The simulated maximum snow depths in 2008 compares quite well with observations for both RIM (0.23 m compared to 0.16 m), and centers (0.39 m compared to 0.46) although the observations show considerable spread (see Nitzbon et al., 2018)." Statements like this are a bit misleading. Should make it clear that the simulated snow depths matching observations is probably mostly good fortune. You are using large-scale forcing from CRU-NCEP. It would be completely unsurprising if the snow depths didn't match up with the observations at the local site when using large-scale forcing. It would be more appropriate to note that due to this good fortune,

it is easier to make direct comparisons to observations.

6. P. 10, line 1: Similar to above, the discrepancy in temperature between model and obs is likely substantially a result of using the large-scale CRUNCEP data to force the model. You wouldn't really expect the soil temperatures to match the observed site level soil temperatures in this circumstance.

7. P. 13, line 4: Same again as above. The stability of the peat plateau is at least partly related to what you are getting from the large-scale forcing. You can't go as far as to make the argument that you have to have certain couplings to maintain the peat plateau permafrost, which is what is implied. What you are finding, which is interesting and important, is just that soil conditions are colder on the peat plateau when snow and water coupling is included.

8. The Discussion section brings up a lot of good points. One thing that isn't clear in the discussion of how one could potentially employ this method at pan-arctic scale is the question of how one would specify the tile structure for each grid cell (is it a polygonal system or a peat plateau, something else, or a mixture of several permafrost landscapes within each large-scale grid cell). Along same lines, how would you know how to initialize the amount and depth of excess ice across the pan-Arctic domain? Based on the information provided in the paper, it seems like this took some trial and error to get it 'right'.

---

## Author Comment (AC1) · 18 Jan 2019

We would like to thank the referee for valuable comments and suggestions our manuscript. Please find our responses and relevant changes (*italic*) to comments (**bold**) below.

**Anonymous Referee #1**

**This paper describes how small-scale surface heterogeneity due to excess ice can be, in a relatively simple way, implemented in land surface models, in this cas, the NOAHMP LSM. A companion paper described a similar work with the Cryogrid model. The motivation of the work is clearly and convincingly laid out, the paper is well structured, easy to read and generally well written (except for frequent systematic grammatical errors). The spirit of the paper is that this work should be seen as a proof of concept, and it is made rather clear that the implementation of such an approach in ESMs will not be an easy task.**

**The methods are described very clearly, and they appear to me appropriate in terms of complexity in the sense that the proposed scheme appears to be on a similar level of complexity as the rest of the model this scheme was implemented in. One might wonder whether some effort should have been devoted to implementing excess ice formation; the possible long time scales involved in the excess ice aggradation could be an argument to discard that option, given that the type of models this approach is designed for is made for centennial-scale simulations, at best.**

We agree that it would be desirable to be able also simulate excess ice formation. However, in addition to the long time scales noted above, the processes behind excess ice formation at the two locations are very different, and a unified method for simulating excess ice formation has therefore not yet revealed itself.

**The discussion of the limitations of this work is honest. I would have liked to see a more thorough analysis of the sensitivity of the model to some critical parameters, in particular those linked to snow; maybe some sensitivity tests might be in order.**

Thank you for this suggestion. We performed a set of simulations to explore the key snow parameter further, which we now describe in the text (section 4.1). See also Fig I.

[Figure]

Figure I: Surface elevation of RIM relative to CENTER at Samoylov Island, Siberia (a), and of PPLAT relative to MIRE at Suossjavri, Northern Norway (b) for different values of $Hs_{min}$.

*An additional set of sensitivity simulations with different values of Hsmin ranging from 0.0 to 0.15 m (not shown) revealed that the landscape evolution at the polygonal site was relatively insensitive to this value, with the transition from LCP to HCP shifting by less than two decades between the minimum and maximum value. A larger sensitivity was seen for the peat plateau site, for which the lowest values of Hsmin resulted in stable permafrost throughout the 21st century.*

**In the discussion (sections 5.1 and 5.2), it would have been good to provide the reader with some more quantitative (if possible) estimates of the importance of neglected processes, and with corresponding priorities in future developments.**

Thank you for this suggestion. We have now added several points to the discussion about limitations, including reference to the snow sensitivity simulations, discussion of the effect of standing water and our opinion on priorities for further developments.

*Simulating surface water in low-centered polygons, or water-filled troughs in the degraded, high-centered stage, would likely modify the results through reduced albedo, increased heat conduction and lower snow redistribution due to smaller elevation differences between the tiles. Results from model simulations, which take larger-scale hydrology into account, show increased soil ALT and earlier permafrost degradation when standing water is included (Langer et al., 2016; Nitzbon et al., 2018).*

*Nevertheless, adding further key processes to the two-tile system is likely to improve the simulation results. Here, we consider the representation of standing water as the most important process, followed by representation of vertically varying organic fractions and soil types, as well as dynamical vegetation. Most of these are already included in several large-scale LSMs (e.g. Lawrence et al., 2011; Reick et al., 2013).*

**Concerning the implementation in ESMs, it is clear that heterogeneity linked to excess ice is relevant only on a small part of the globe. In many other places, the most relevant heterogeneities are linked to vegetation, orography, or other factors. Can the authors think of a more general (globally relevant) tiling concept in which the tiling linked to excess ice could be integrated?**

This is a good point. We have added the following on this topic.

*Regardless of the choice of implementation, the method proposed here should be considered in the context of a larger effort to improve the representation of horizontal land processes ESMs. The land component of coupled atmosphere-land surface models is typically of considerable complexity in the vertical dimension, but includes little horizontal interaction and variation (see e.g. Clark et al., 2015). While representing heterogeneous excess ice is a relevant only in certain regions, we believe that a more flexible model structure with individual sub-grid soil columns that can exchange water and snow is a concept that deserves further investigation also in other regions.*

**In conclusion, I definitely think that this paper should be published if the points above and some specific points mentioned below are addressed. This should only require minor modifications.**

**Specific points:**

**P5 L28: FEXICE (and similar variable names in the text): In the figure, you use FEXICE (EXICE as an index), so please do so in the main text, too.**

Done!

**P6 L27: In the equation and in the text, Ksat is a constant. Call it Ksat,0 for Ksat at the surface, to prevent confusion.**

Corrected. Thank you!

**P7 L10: "The lateral ground heat flux [. . .] between two grid cells with overlapping soil depth. . ." Cells or tiles? Probably tiles.**

This should be *tiles*, as you assumed. Corrected.

**P7 L17-18: Not clear why the elevated tile is used as a reference. In most models, there is no excess ice yet, so it might have been more appropriate to use the lower tile as a reference (especially because you do not use stagnant water at the surface anyway).**

By using the elevated tile as reference, we use Lee et al. (2014) as the starting point. This is now stated in the text. Although it is not common to include excess ice yet, this is more useful in terms of evaluating the effect of the different lateral fluxes.

*At both locations, a separate reference simulation (REF) is run with the same initial conditions as the elevated tile in the laterally coupled system (RIM or PPLAT), corresponding to the same model setup as employed in Lee et al. (2014), i.e. a 1D excess ice representation without lateral exchange.*

**By the way, it would have been nice, in the discussion, to sped a few lines on discussing how taking into account stagnant water could have changes the results. In my opinion, it could have very major impacts.**

See response to comment below (P.15L23).

**P7 L29: "This expands the soil thickness of the RIM with 1.5 m". Wouldn't "by 1.5 m" be better English?**

Yes. This has now been corrected.

**P8 L2: "we additionally add excess ice to the bottom soil layer (in both coupled tiles)": In the figure it looks like the 35 cm excess ice are added to the lowest layer only in the lower tile. Please clarify.**

Excess ice was added to both tiles, but with unequal amounts. This has now been clarified:

*To allow the RIM to sink below the elevation of the center, we add excess ice to the bottom soil layer, with the largest amount in CENTER, so that the total elevation difference is only 35 cm (Fig. A1). This is an approximate average value for observed rim heights at Samoylov.*

**P8 L14: "but still show continued": -> shows. In many places, there are wrong or missing s's (wrong plurals, wrong conjugation). Please go through the text carefully.**

Thank you for pointing this out. We have now corrected this and other similar errors.

**P8 L14: "making which makes"**

Corrected!

**P9 L2: I understand why you introduce figure 7 here before figures 5 and 6, but I think that the figure numbers should be in order of appearance in the text nevertheless.**

Agreed. We have now removed the reference to fig. 7 here so that the figure numbers agree with the order of appearance in the text.

**P9 L20: "simulation.." (only one point needed) Same line: "become is": ?**

Both corrected. Thank you!

**P10 L19: Why do you call the sensible heat flux HFX? Doesn't make much sense to me.**

This is the name of this variable in the NoahMP model, but we agree that this is not intuitive and have changed the name to the more commonly used *SH*.

**P11 L23: Replace whereas by while (I think)**

Changed. Thank you!

**P12 L16: scarcely -> barely?**

Changed. Thank you!

**P13 L26: "becoming in equilibrium": Are you sure that this is good English?**

We agree that this was not a good expression. This has now been rephrased:

*quickly reaching equilibrium*

**P14 L3: replace instantaneously by instantaneous (and does by do on the same line)**

Done.

**P14 L27: As said before, a sensitivity test showing the effect of the snow parameters would have been interesting. Or would that be too model-specific?**

See reply to comment above, including Fig. I.

**P15 L23: "Simulating instead surface water in low-centered polygons, or waterfilled troughs in the degraded, high-centered stage, would modify the results presented here." As said before, I'd like to see a discussion how this would modify the results (in your expert opinion)**

This has now been included, based on results from the Cryogrid model:

*Simulating surface water in low-centered polygons, or water-filled troughs in the degraded, high-centered stage, would likely modify the results through reduced albedo, increased heat conduction and lower snow redistribution due to smaller elevation differences between the tiles. Results from model simulations, which take larger-scale hydrology into account, show increased soil ALT and earlier permafrost degradation when standing water is included (Langer et al., 2016; Nitzbon et al., 2018).*

---

## Author Comment (AC2) · 18 Jan 2019

We would like to thank the referee for valuable comments and suggestions our manuscript. Please find our responses and relevant changes (*italic*) to comments (**bold**) below.

**Anonymous Referee #2**

**In this paper, the authors take steps towards an ability to represent in a large-scale model the important lateral snow redistribution, water, and heat processes that impact the trajectory of permafrost thaw and related processes in different permafrost landscapes. The approach is parsimonious, which I like. The authors propose to represent these systems with just two 'tiles' (rim and center for polygonal tundra), rather than explicitly modeling the full complexity of the heterogeneous landscape. I like this approach as it does lend itself to potential inclusion across the pan-Arctic. A significant limitation is that the model is not explicitly modeling the formation of these permafrost landscape features. Instead, the goal is simply to be able to simulate the transition from a low-centered to a high-centered polygon. This is a reasonable first step and the authors acknowledge this limitation. Clearly, to have 'full' confidence in the model, one would want it to be able to simulate the full set of physical processes that drive both the formation and the decay of low-centered polygons. Nonetheless, this is a practical first step that is clearly an improvement over the current 1-tile assumption that cannot at all account for the real spatial heterogeneity of the system.**

As noted also in the reply to referee #1, we completely agree that simulating the formation of excess ice would be desirable, although we do not see this as feasible within the current study, both due to the long time scales, and the complexity and lack of well-developed parameterizations for the buildup processes.

**Overall, I enjoyed reading this paper and I find it suitable for publication with a few relatively minor revisions and clarifications.**

**Specific comments**

**1. When the Noah-MP model is introduced, it would be good to explain why Noah-MP is being used instead of any other model. I believe that it is because of the lateral flow capabilities in WRF-hydro, but that capability isn't introduced until section 2.2.4.**

We now include a short justification for the use of this model section 2.2, when the NoahMP model is first introduced.

*Furthermore, lateral subsurface water fluxes are already implemented in this model as part of the WRF-Hydro modelling system (see sec. 2.2.4). With some modifications it is therefore a suitable base model for studying the geophysical aspects of permafrost thaw, including the importance of lateral fluxes.*

**2. P. 8, line 2 typo: "only elevated only"**

Corrected. Thank you!

**3. I wonder if the "coupled" is the best way to reference the multiple tile simulations. Coupled can mean a lot of things in different contexts. Perhaps you could rename as Reference and Tiled or Single column and Two column or something else that is more descriptive.**

We agree that only referring to the two-tiled simulation as the "coupled" simulation is ambiguous. We have now carefully gone through the manuscript to make sure that whenever we refer to the "coupled" simulation, it is clear that we are referring to lateral coupling between tiles.

**4. Figure 5: Why is the ref simulation at depth so much warmer than either the RIM or CENTER simulation?**

We attribute this to the non-linear effect of snow. Maintaining an almost snow-free rim throughout the winter season increases the heat loss more on the RIM than it is reduced from the CENTER. The tiled system is therefore colder than the REF which receives the average snow accumulation.

**5. P.9, Line 16: "The simulated maximum snow depths in 2008 compares quite well with observations for both RIM (0.23 m compared to 0.16 m), and centers (0.39 m compared to 0.46) although the observations show considerable spread (see Nitzbon et al., 2018)." Statements like this are a bit misleading. Should make it clear that the simulated snow depths matching observations is probably mostly good fortune. You are using large-scale forcing from CRU-NCEP. It would be completely unsurprising if the snow depths didn't match up with the observations at the local site when using large-scale forcing. It would be more appropriate to note that due to this good fortune, it is easier to make direct comparisons to observations.**

We agree that the raw CRU-NCEP data cannot be expected to reproduce local snow depths accurately, and the agreement is partly due to the scaling factor for precipitation. This is now noted in the text:

*This was partly achieved by applying a scaling factor for precipitation (Pscale) of 0.6 (Table 2).*

**6. P. 10, line 1: Similar to above, the discrepancy in temperature between model and obs is likely substantially a result of using the large-scale CRUNCEP data to force the model. You wouldn't really expect the soil temperatures to match the observed site level soil temperatures in this circumstance.**

We again agree that one cannot expect to match soil temperatures exactly when forced with a large-scale reanalysis like CRU-NCEP. This is now pointed out in the discussion section.

*However, given the relatively coarse resolution of the forcing data, a certain disagreement must be expected*

**7. P. 13, line 4: Same again as above. The stability of the peat plateau is at least partly related to what you are getting from the large-scale forcing. You can't go as far as to make the argument that you have to have certain couplings to maintain the peat plateau permafrost, which is what is implied. What you are finding, which is interesting and important, is just that soil conditions are colder on the peat plateau when snow and water coupling is included.**

We agree that permafrost could be maintained without these couplings in colder conditions. However, the snow and soil water conditions are recognized also by others as key factors for maintaining these marginal permafrost features in this region, which we now also include a reference for.

*This is in agreement with previous studies of palsas and peat plateaus in this region, pointing to low snow accumulation and dry peat during summer as the most important factors for their stability (see Seppälä, 2011).*

**8. The Discussion section brings up a lot of good points. One thing that isn't clear in the discussion of how one could potentially employ this method at pan-arctic scale is the question of how one would specify the tile structure for each grid cell (is it a polygonal system or a peat plateau, something else, or a mixture of several permafrost landscapes within each large-scale grid cell). Along same lines, how would you know how to initialize the amount and depth of excess ice across the pan-Arctic domain? Based on the information provided in the paper, it seems like this took some trial and error to get it 'right'.**

This is a good point. We have expanded the discussion with some more details on this:

*Ground ice data from Brown et al. (1998) could provide a starting point here, similar to the study by Lee et al. (2016).Assigning excess ground ice to the first soil layers below the simulated ALT has been a reasonable first-order choice for the two test sites, but this procedure is likely not adequate for areas with excess ice well below the current active layer, e.g. due to burial or melting of excess ground ice in the past (e.g. truncated ice wedges, Brown, 1967). Ultimately, new global data sets for ground ice depth, excess ice density and geometries of the two tiles must be compiled, for example building on approaches as in Hugelius et al, (2014) and Strauss et al., (2017).*